# Social and structural determinants of injecting-related bacterial and fungal infections among people who inject drugs: protocol for a mixed studies systematic review

Thomas D Brothers [1,2] Dan Lewer [1] Matthew Bonn [3] Duncan Webster [2,4] Magdalena Harris [5]

For numbered affiliations see end of article.

**Correspondence to**
Dr Thomas D Brothers;
thomas.brothers@dal.ca

## ABSTRACT

**Introduction** Injecting-related bacterial and fungal infections are a common complication among people who inject drugs (PWID), associated with significant morbidity and mortality. Invasive infections, including infective endocarditis, appear to be increasing in incidence. To date, preventive efforts have focused on modifying individual-level risk behaviours (eg, hand-washing and skin-cleaning) without much success in reducing the population-level impact of these infections. Learning from successes in HIV prevention, there may be great value in looking beyond individual-level risk behaviours to the social determinants of health. Specifically, the risk environment conceptual framework identifies how social, physical, economic and political environmental factors facilitate and constrain individual behaviour, and therefore influence health outcomes. Understanding the social and structural determinants of injecting-related bacterial and fungal infections could help to identify new targets for prevention efforts in the face of increasing incidence of severe disease.

**Methods and analysis** This is a protocol for a systematic review. We will review studies of PWID and investigate associations between risk factors (both individual-level and social/structural-level) and the incidence of hospitalisation or death due to injecting-related bacterial infections (skin and soft-tissue infections, bacteraemia, infective endocarditis, osteomyelitis, septic arthritis, epidural abscess and others). We will include quantitative, qualitative and mixed methods studies. Using directed content analysis, we will code risk factors for these infection-related outcomes according to their contributions to the risk environment in type (social, physical, economic or political) and level (microenvironmental or macroenvironmental). We will also code and present risk factors at each stage in the process of drug acquisition, preparation, injection, superficial infection care, severe infection care or hospitalisation, and outcomes after infection or hospital discharge.

**Ethics and dissemination** As an analysis of the published literature, no ethics approval is required. The findings will inform a research agenda to develop and implement social/structural interventions aimed at reducing the burden of disease.

### Strengths and limitations of this study

► Injecting-related bacterial and fungal infections are a common problem among people who inject drugs (PWID), associated with significant morbidity and mortality.

► Understanding the social and structural determinants of injecting-related bacterial and fungal infections could help to identify new targets for prevention efforts, in the face of increasing incidence of severe disease.

► People with lived and living expertise of injection drug use, as well as practicing clinicians caring for PWID with injecting-related infections, and experienced qualitative, quantitative and mixed methods researchers, were involved in the development of the protocol and will participate in analysis and synthesis.

► Reviewing quantitative, qualitative and mixed methods studies enables a broad scope and the inclusion of important contextual information that may not be captured in quantitative studies alone.

► This review only includes studies with infection-related outcomes, and so may not include some studies with risk factors for infection (eg, hand-washing and skin-cleaning) as outcomes.

**PROSPERO registration number** CRD42021231411.

## INTRODUCTION

Injecting-related bacterial infections are very common among people who inject drugs (PWID), with up to 39% of PWID reporting skin infections within the past month and 70% reported lifetime prevalence.[1] If not effectively treated, superficial infections can progress and spread through the bloodstream, leading to life-threatening infections such as infective endocarditis, osteomyelitis, septic arthritis or epidural abscess.[2] Pathogenic bacteria or fungi may be introduced directly

into the bloodstream as endogenous skin flora, or from non-sterile injecting equipment or contaminated drugs.[3] As a result, PWID face extremely high risk of disability or death from severe bacterial and fungal infections that require hospitalisation for antibiotics and/or surgery.[4–9] The incidence of severe bacterial and fungal infections is increasing among PWID in Canada,[10–12] the United Kingdom,[13] the USA,[14–19] Australia[20] and Sweden.[21] A recent modelling study suggests that one in five PWID in the USA may die of infective endocarditis alone over the next 10 years.[19] Better approaches to prevention are urgently needed.

To date, prevention efforts for injecting-related bacterial and fungal infections have primarily focused on changing individual-level risk behaviours,[1 22] including hand-washing before drug preparation,[23] skin-cleaning before injecting[24] and avoiding subcutaneous or intramuscular injections.[25] Although individual-level interventions may be effective for people who due to their socioeconomic situation can adopt these practices,[26 27] they have not translated into reductions in the population-level incidence of severe bacterial and fungal infections, which continue to rise.

An appreciation for the importance of social and structural determinants of health has emerged from the understanding that individual health behaviours are shaped by contextual factors.[28–32] The 'risk environment' conceptual framework, as developed by Rhodes *et al*,[33–36] describes how interactions between social and structural factors external to the individual influence individual behaviours, and therefore structure or create health harms.[37] The risk environment framework has informed clinical and public health efforts at reducing other drug-related harms, including HIV transmission,[33 36 38] hepatitis C virus (HCV) treatment[39] and drug overdoses.[29 40] Better understanding of the social determinants of health can help to shift beliefs about responsibility and risk from individual behaviours to the places and social situations in which individuals exist, and inform the development of innovative interventions addressing both social and individual-level factors.[36 41]

Like other drug-related harms, risk for injecting-related bacterial and fungal infections also likely reflects contributions of multiple, interacting factors external to individuals that influence risk behaviours and therefore shape health outcomes. For example, homelessness may constrain an individual's ability to wash their hands or use sterile water for injecting,[42] and policy constraints on needle and syringe programs (from criminalisation to reduced operating hours) create a situation in which an individual is more likely to reuse a blunted or contaminated needle. Stigma and criminalisation of people who use drugs may keep people away from primary healthcare, causing superficial bacterial infections to remain untreated and progress to enter the bloodstream. In response to the alarming increases in incidence, calls to enhance understanding of the social determinants of injecting-related bacterial and fungal infections have

emerged from both PWID[43] and academic communities.[2 4 26 44] Learning from the successes of HIV-prevention efforts in particular, mapping the risk environment for injecting-related bacterial and fungal infections could inform new prevention efforts that target social and structural causes.[45 46]

## Objectives

To better understand the risk for injecting-related bacterial and fungal infections among PWID, we seek to conduct a systematic mixed studies review (of quantitative, qualitative and mixed methods evidence) to identify social and structural determinants of these infections among PWID, informed by the risk environment framework. The proposed review will seek to answer the question, 'Among people who inject drugs (PWID), what social and structural factors are associated with the development, treatment, and outcomes of injecting-related bacterial and fungal infections?'

## METHODS AND ANALYSIS

This systematic review protocol follows the Preferred Reporting Items for Systematic Review and Meta-Analysis Protocols guideline[47 48] and is informed by guidance on conducting systematic reviews of association (aetiology) by the Joanna Briggs Institute[49] and on conducting and reporting mixed studies reviews by Pluye and colleagues[50 51] and by Tricco and colleagues.[52] We plan to conduct this study over 12 months following the initial full database search on 18 February 2021.

'Mixed studies reviews' incorporate synthesis of quantitative, qualitative and mixed methods studies; they are one of the several emerging methods for systematically incorporating evidence from diverse methodologies (eg, integrative review, realist review, meta-narrative review and critical interpretive synthesis).[50 52 53] Mixed studies reviews use a mixed methods approach.[54 55] They are particularly useful for research questions like ours that require detailed contextual information available from qualitative and mixed methods studies, and for synthesis plans like ours that incorporate a qualitative content analysis directed by existing theories and conceptual frameworks.[50 52]

We have searched the PROSPERO registry of systematic reviews and found none in progress related to our objectives.

## Conceptual framework

The risk environment framework is the most prominent social–ecological model for substance use research; it comprises risk factors external to individuals, considering types and levels of environmental influence.[34 35 37 56] As first developed in the context of HIV prevention, the risk environment framework describes four different types of environmental influences: social, physical, economic and policy. These can occur at two different levels, microenvironmental or macroenvironmental.[38]

Microenvironmental factors operate at the level of interpersonal relationships, community and group norms, and institutional or organisational responses.[36] This could include local norms about the culture of substance use and acceptability of receptive sharing or reuse of potentially contaminated injecting equipment among PWID (a social factor), or increasing housing prices (an economic factor) contributing to homelessness and lack of access to soap and water (physical factors). Macroenvironmental factors operate at the level of states, societies, and laws, and interact with microenvironmental factors.[36] This could include state policing crackdowns on heroin importation leading instead to increased importation of fentanyl (a policy factor), which has a shorter half-life and associated risks of increased injecting frequency.[57]

## Eligibility criteria

Informed by the Population, Exposures, Outcomes approach,[49] we will include peer-reviewed papers describing eligible studies according to the following criteria.

### Study designs

We will include studies measuring quantitative associations between exposures and outcomes of interest (as described below), and studies reporting qualitative data on relationships between experiences of exposures and outcomes. We will exclude case-reports and case series that do not include analyses of association, and we will exclude reviews, commentaries and editorials, as they do not include the original data.

### Participants

We will include studies examining PWID of any age or nationality. By PWID, we are referring to people who inject opioids (eg, heroin, fentanyl, hydromorphone and morphine), stimulants (eg, cocaine and methamphetamines) or other psychoactive substances via intravenous, intramuscular, or subcutaneous routes. We expect there to be differences between studies in how PWID are operationally defined (eg, ever injected or recently injected in past year or past 3 months).[1] Taking a broad perspective, we will include studies that identify participants as PWID at risk for injecting-related infections, and not limit this to self-reported current or recent injecting. Many PWID transition back and forth between injection and non-injection routes of consumption,[58 59] and PWID who have stopped injecting may experience persistently increased risk for invasive infections such as endocarditis, due to prior damage to vasculature and heart valves.[2] Studies that focus only on people who inject performance-enhancing drugs (eg, anabolic steroids) or gender-affirming hormones will be excluded as these are not psychoactive substances.

### Exposures

We will include studies addressing risk factors or determinants for injecting-related bacterial and fungal infections among PWID, taking a broad perspective. In addition to studies that directly survey individual participants about potential risk factors, we will consider studies investigating potential associations with social or structural risk factors at microenvironmental and macroenvironmental levels. We will also include studies of interventions that aim to reduce risk, including educational, behavioural or structural (eg, community mobilisation or policy change) interventions.[26 27 46] Studies that only report on individual-level risk factors (eg, skin-cleaning) will be included in the search, and then classified separately in the data synthesis.

### Outcomes

Outcomes of interest include incidence of, hospitalisation with, or mortality due to, injecting-related bacterial and fungal infections. Bacterial and fungal infections include skin and soft-tissue infections (cellulitis, abscess, necrotising fasciitis), bloodstream infections (bacteraemia), vascular infections (endocarditis, septic or suppurative phlebitis), bone and joint infections (osteomyelitis, septic arthritis or discitis) and central nervous system infections (epidural abscess, brain abscess, meningitis or encephalitis). We will also include as outcomes, diseases caused by pathogenic bacteria and fungi associated with drug supply, preparation and injecting practices, including tetanus,[22] botulism,[3] invasive group A streptococcal disease,[60] as well as methicillin-sensitive *Staphylococcus aureus* and methicillin-resistant *Staphylococcus aureus* (MRSA).[61] We will exclude studies identifying only non-injecting-related bacterial and fungal infections as outcomes, such as tuberculosis, pneumonia, chlamydia or gonorrhoea, and bacterial infections that are primarily sexually transmitted, for example, syphilis.

We will extract all outcomes as reported. We expect there to be many different methods for identifying these outcomes, including participant self-report of infection or hospitalisation, administrative records of hospital admission and mortality records from vital statistics.

### Time frame, setting and language

We will include studies published between 1 January 2000, and the planned search date, 18 February 2021, to capture contemporary research that would be more likely to inform policy and clinical practice. There will be no restrictions by study setting. We will include articles in English and French. A list of potentially relevant titles reported in other languages will be provided as an online supplemental appendix.

### Information sources

We will search PubMed, EMBASE, Scopus, CINAHL and PsycINFO databases. Electronic database searches will be supplemented by manually reviewing reference lists of included studies, and forward 'snowball' searching by identifying articles that cite included studies.[62] We will also circulate a bibliography of the included articles to the systematic review team, which includes people with lived (past) and living (current) expertise of injection drug

use and clinicians who care for people with injection-associated infections. We will include articles identified from the personal files of the systematic review team that were not identified in the bibliography.

## Search strategy

Our search strategy is informed by several key existing systematic reviews or review protocols. A 2017 systematic review identified prevalence and individual-level risk factors for injecting-related injuries and diseases among PWID, including bacterial infections.[1] Previous systematic reviews have applied the risk environment framework to understand HIV transmission,[33] HIV[63] and HCV[39] treatment access, HIV disease progression,[64] providing or receiving assistance with the transition to injection drug use,[65] and 'safer environment interventions'[41] among PWID. Two published protocols of systematic reviews in progress focused on health services and interventions to treat or prevent viral and bacterial infections among PWID.[66 67]

The final search strategy will be developed by our review team in consultation with health information specialist-librarians with systematic review experience. We conducted a pilot search in PubMed and validated by checking for inclusion of key, recent studies known to the authors that assessed either risk factors for severe bacterial or fungal infections among PWID,[68 69] social/structural determinants of bacterial infections among PWID[42 70–73] or complex interventions to reduce risk of bacterial infections among PWID[26] (online supplemental appendix table 1). This validation process led to several iterative changes, including specific search terms related to 'acidifiers' and 'groin injecting'. See table 1 for the draft search strategy in PubMed, which identified 1752 potentially relevant sources on 1 February 2021.

## Data management and reference selection

Search results will be uploaded into Covidence, a cloud-based software programme, where they can be automatically checked for de-duplication. Two reviewers will screen titles and abstracts against the inclusion criteria. We will obtain full text reports for all titles that appear to meet the inclusion criteria, or for where there is uncertainty. Two reviewers will screen these full text reports and mark each as included or excluded, recording reasons for exclusion. We will resolve discrepancies through discussion.

## Data collection process

The team will develop and pilot-test a data extraction form, which can be applied in Covidence. We will extract information on the following:

- ► Study date (or publication date, if study date unavailable).
- ► Study country/region.
- ► Methodology (quantitative, qualitative or mixed methods).
- ► Study design.
- ► Sample size and demographic characteristics (age, gender, race/ethnicity).
- ► Inclusion criteria and definition of PWID.
- ► Sampling methods and recruitment setting (eg, community recruitment, needle exchange clients, clinic, hospital).
- ► Outcomes reported (incident infection/diagnosis, clinic visit, emergency department visit, hospital admission or death).
- ► Outcome operational criteria (self-reported, medical record review and administrative data).
- ► Infections reported (skin and soft-tissue infections at injection sites (e.g. abscess, cellulitis), bacteraemia, infective endocarditis, osteomyelitis, septic arthritis or other).[1]
- ► Exposures, risk factors or interventions assessed (including operational criteria).
- ► Measures of association between risk factors and outcomes (for quantitative and mixed methods studies).
- ► Summary of findings.
- ► Implications for policy and/or practice, as reported by authors.[66]
- ► Implications for research, as reported.
- ► Gaps identified, as reported.

## Critical appraisal/risk of bias assessment

We will apply a formal, validated critical appraisal tool for mixed studies reviews, the Mixed Methods Appraisal Tool, 2018 edition, which is designed for use with quantitative, qualitative and mixed methods studies.[74 74 75] It includes the following five core quality criteria for five categories of study designs: (a) qualitative, (b) randomised controlled/interventional, (c) non-randomised controlled, (d) quantitative descriptive and (e) mixed methods. This tool does not assign an overall quality score; rather, we will consider

| Table 1 | A model of the risk environment tabular summary | |
|---|---|---|
| | **Microenvironmental** | **Macroenvironmental** |
| Social environment | | |
| Physical environment | | |
| Economic environment | | |
| Policy environment | | |

Modified from the study by Rhodes.[34]

each item in the risk of bias assessment independently. Owing to the broad scope of this review, we will not exclude studies deemed low quality. Study quality and risk of bias will inform interpretation of the data.

## Data synthesis and analysis

We will perform data synthesis and analysis over several stages. We will perform a quantitative tabular synthesis of included studies, by year; country/region; methodology (quantitative, qualitative or mixed methods), infection type (cellulitis, endocarditis, etc.) and outcome (incident infection, care of superficial infection, care of severe infection/hospitalisation or outcomes after infection).

We will then conduct a qualitative, directed content analysis, as described by Hsieh and Shannon,[76] to map the risk environment for injecting-related bacterial and fungal infections. We will first code each included exposure or risk factor according to the risk environment framework as individual-level or social/structural. Social/structural factors are those that are external to the individual, and together make up the risk environment. We will then code each social/structural factor according to the type (social, economic, political or physical) and level (macroenvironmental or microenvironmental) of the risk environment. We will use a hybrid deductive–inductive approach, as described by Pluye and Hong[50]; we will start with the existing risk environment conceptual framework but allow for new concepts related to the classification of specific factors to emerge. To improve reliability, coding will be performed in tandem by a reviewer pair for three studies, with discrepancies resolved through discussion.

Coding will then be discussed with the whole review team to identify improvements to the coding approach and framework. The remaining studies will be coded by one reviewer. We will present a tabular synthesis categorising each social/structural factor according to type and level in the risk environment framework,[49] and a narrative synthesis of the existing evidence for each social/structural factor.[39] See table 1 for the model table framework.

To explore how different macroenvironmental and microenvironmental factors influence individual-level risk behaviours, we will present a narrative synthesis of identified individual-level and social/structural factors affecting each step of a potential pathway from (a) drug acquisition; (b) drug preparation; (c) drug injection; (d) development of and care for superficial infections (eg, self-treatment or primary/emergency department care); (e) development of and care for severe or invasive infections (eg, self-care or hospital care) and (f) outcomes after infection or hospital discharge, including infection-related mortality.[2] See figures 1 and 2 for illustrative schematics of how macroenvironmental and microenvironmental factors influence individual-level factors and may increase risks of injecting-related bacterial infections at two different stages in the pathway from drug acquisition to outcomes after severe/invasive infections. We plan to develop and refine these figures to reflect findings of the review. Note that because this review includes only studies with infection-related outcomes, we may not identify or include all published studies that identify determinants of intermediate risk factors or behaviours;

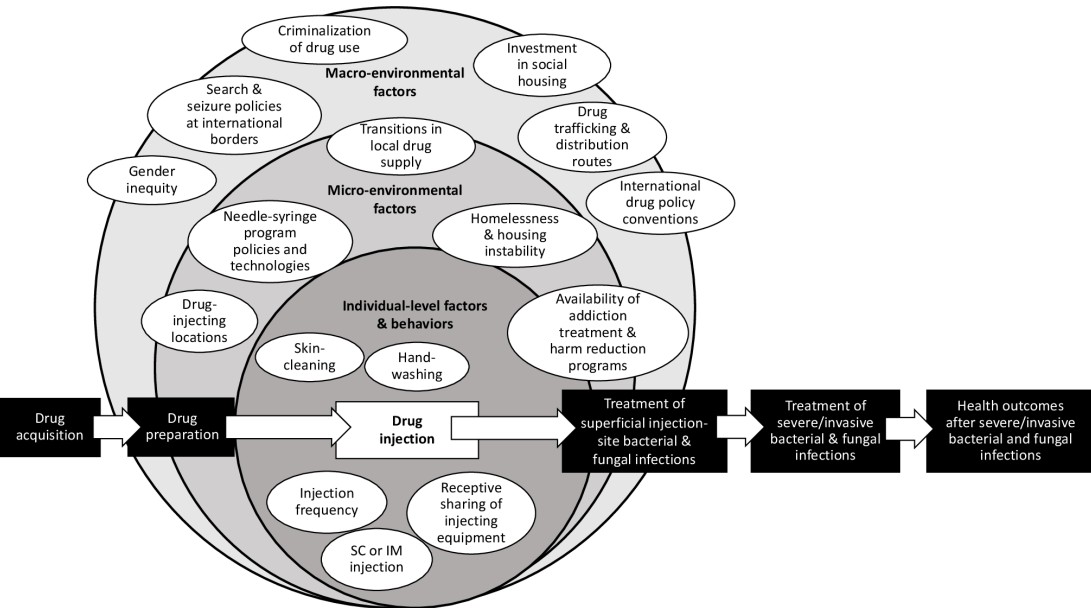

**Figure 1** Illustrative schematic of a potential risk environment for injecting-related bacterial and fungal infections, as it may structure or create infection risk during the process of drug injection/consumption. Environmental factors, which are external to individuals, interact to influence individual-level factors and health behaviours across stages of a potential pathway: drug acquisition; drug preparation; drug injection; treatment of superficial injection-site bacterial and fungal infections (eg, in primary care or emergency departments); treatment of severe/invasive bacterial and fungal infections (eg, in hospital for intravenous antibiotics and/or surgery) and health outcomes after severe/invasive bacterial and fungal infections (eg, disability, death). IM, intramuscular; SC, subcutaneous.

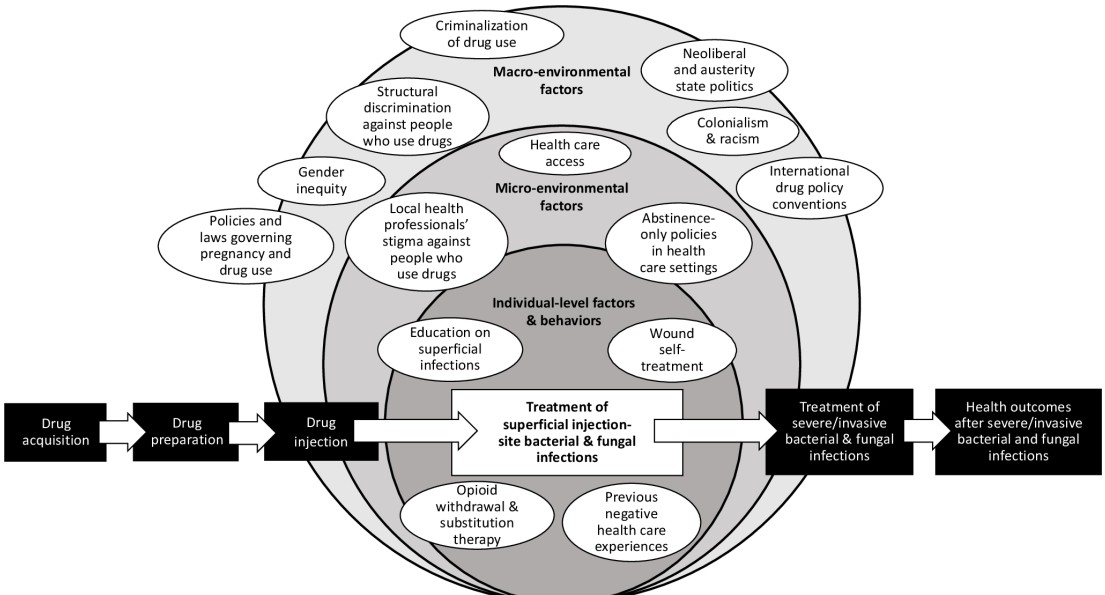

**Figure 2** Illustrative schematic of a potential risk environment for injecting-related bacterial and fungal infections, as it may structure or create infection risk during the process of recognition and adequate treatment of superficial injection-site infections. Environmental factors, which are external to individuals, interact to influence individual-level factors and health behaviours across stages of a potential pathway: drug acquisition; drug preparation; drug injection; treatment of superficial injection-site bacterial and fungal infections (eg, in primary care or emergency departments); treatment of severe/invasive bacterial and fungal infections (eg, in hospital for intravenous antibiotics and/or surgery) and health outcomes after severe/invasive bacterial and fungal infections (eg, disability, death).

for example, some studies identifying determinants of skin cleaning before injecting may not be included in this review if they do not consider infection-related outcomes.

If we identify sufficient quantitative data, we will conduct a meta-analysis of summary effects of association between social/structural factors (eg, homelessness) and injecting-related infection outcomes among PWID (eg, hospitalisation with any severe injecting-related bacterial or fungal infection) identified in our content analysis. Where available, we will extract measures of association adjusted for confounding, including point estimates and standard errors. To assess feasibility, we will evaluate clinical and methodological heterogeneity by comparing study design, sample characteristics and operational definitions of exposures and outcome.[65 77] We will measure statistical variability between studies using $I^2$ statistics.[65 77 78]

### Patient and public involvement
The research team includes people with lived/living expertise of injection drug use and clinicians caring for PWID with bacterial and fungal infections. The research questions and analysis approach have been designed collaboratively, and build on existing relationships between authors with lived/living expertise and academic/medical expertise.[79–81] The topic was inspired by the deaths of friends and patients to injecting-related bacterial infections, and a recognition of the need to look broader than individual-level interventions.[82–84]

### ETHICS AND DISSEMINATION
As this is a secondary analysis of published literature, no ethics approval is required.

By mapping the risk environment for injecting-related bacterial and fungal infections among PWID, this study aims to identify opportunities for new policy and practice approaches to prevention. This may include highlighting the importance of scaling up access to existing interventions (eg, needle/syringe distribution; opioid agonist treatment) or identifying opportunities for novel combined social and clinical interventions (eg, providing Housing First along with antibiotics and addiction treatment for PWID hospitalised with invasive infections). Our evidence synthesis may identify opportunities for 'structural interventions', which promote health by altering the social–structural context which influence health.[46 85] Examples include the potential impacts of policy or law changes (eg, decriminalising substance use may combat stigma and facilitate access to primary care of injecting-related infections) or of community mobilisation (eg, organised PWID empowered to effectively demand safe and welcoming hospital care).[44 86 87]

Incidence of HIV and HCV infections are increasing in parallel with injecting-related bacterial and fungal infections among PWID, in the context of the North American overdose crisis. Although our review focuses only on injecting-related bacterial and fungal infections, we expect to find that these outcomes share many overlapping social–structural correlates.[29 45 79 85 88] Secular increases in drug-related infections and overdoses over

the past decade have been describes as a 'syndemic', with shared, fundamental causes rooted in the 'War on Drugs' and austerity politics.[29 42 79 85] Accordingly, findings from our study may be able to inform risk reduction interventions for multiple health outcomes among PWID. The COVID-19 pandemic has accelerated the overdose death crisis, deepened poverty and marginalisation, and interrupted access to harm-reduction services[79 85]—the consequential impacts on injecting-related bacterial and fungal infections, and potential for specific interventions, is not yet clear.

The risk environment approach may also help to shift beliefs regarding responsibility and risk from individual behaviours to social and structural causes, which may change attitudes of healthcare professionals and systems regarding treatment of severe infections, including procedures such as heart valve replacement surgeries for injecting-related infective endocarditis.[89–91] This may also inform educational and behaviour-change interventions, highlighting the importance of addressing more immediate, pragmatic priorities (eg, housing and income, or vein access and care) that may also have benefits in infection risk reduction.[92] Extending further concepts from the HIV prevention social science literature, we are also interested in how our findings may relate to interventions in social networks.[93] Social relationships within networks of PWID influence the likelihood of individual risk or protective behaviours (eg, needle re-use; skin cleaning). Although bacterial and fungal infections are generally not transmitted through blood, as is HIV or HCV, PWID in the same network may experience transmission of colonisation with pathogenic bacteria (eg, MRSA)[94] or heightened risk of accessing a shared, contaminated drug supply (eg, outbreaks of botulism[3 95] or anthrax[96]).

By emphasising the potential importance of social and structural factors, the findings will also inform future epidemiological research using large-scale linked administrative, data linking individual and clinical risk factors with information on social factors.[69 97] Knowledge translation activities will include peer-reviewed academic publications and conference presentations, meetings with drug user groups/unions and harm reduction programmes, and also communications to the lay public in targeted outlets.

**Author affiliations**
[1]UCL Collaborative Centre for Inclusion Health, Institue of Epidemiology and Health Care, University College London, London, UK
[2]Department of Medicine, Dalhousie University, Halifax, Nova Scotia, Canada
[3]Canadian Association of People Who Use Drugs, Dartmouth, Nova Scotia, Canada
[4]Division of Infectious Diseases, Saint John Regional Hospital, Saint John, New Brunswick, Canada
[5]Department of Public Health, Environments and Society, London School of Hygiene & Tropical Medicine, London, UK

**Acknowledgements** We appreciate the contributions to science of all participants in the included studies. TDB, MB and DW live and work in Mi'kma'ki and along the Wolastoq, the ancestral and unceded territory of the Mi'kmaq and the Wolastoqiyik. We are all Treaty people. We thank Prof Andrew Hayward for helpful suggestions on the synthesis and analysis plan. We thank the reviewers and editor for helpful suggestions that improved the manuscript.

**Contributors** This study was conceived by TDB, DL, MB, DW and MH. The pilot search strategy was developed by TDB, and revised with input from DL and MB. TDB, DL and MB will be involved in the data collection for this study. All authors will be involved in data interpretation and analysis. TDB wrote the first draft of this manuscript. All authors provided critical feedback and intellectual input. All authors provided their final approval for the publication of this version of the manuscript. As guarantor, TDB accepts full responsibility for the work and controlled the decision to publish.

**Funding** TDB is supported by the Dalhousie University Internal Medicine Research Foundation Fellowship, Killam Postgraduate Scholarship, Ross Stewart Smith Memorial Fellowship in Medical Research and Clinician Investigator Programme Graduate Stipend (all from Dalhousie University Faculty of Medicine), a Canadian Institutes of Health Research Fellowship (CIHR-FRN# 171 259), and through the Research in Addiction Medicine Scholars (RAMS) Programme (National Institutes of Health/National Institute on Drug Abuse; R25DA033211). DL is funded by a National Institute of Health Research Doctoral Research Fellowship (DRF-2018–11-ST2-016). MB was supported in this work via the Ross Stewart Smith Memorial Fellowship in Medical Research, from Dalhousie University Faculty of Medicine. MH is funded by a National Institute of Health Research Career Development Fellowship (CDF-2016-09-014).

**Disclaimer** The views expressed are those of the author(s) and not necessarily those of the NHS, the NIHR or the Department of Health and Social Care. These funders had no role in the conduct or reporting of the research.

**Competing interests** MB reports personal fees from AbbVie, a pharmaceutical research and development company, and grants and personal fees from Gilead Sciences, a research-based biopharmaceutical company, outside of the submitted work.

**Patient consent for publication** Not required.

**Provenance and peer review** Not commissioned; externally peer reviewed.

**ORCID iDs**
Thomas D Brothers http://orcid.org/0000-0002-5570-5556
Dan Lewer http://orcid.org/0000-0003-3698-7196
Matthew Bonn http://orcid.org/0000-0002-6406-0171
Duncan Webster http://orcid.org/0000-0001-7692-7150
Magdalena Harris http://orcid.org/0000-0001-8718-8226

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
