## [Reviewer comments · BMJ Open]

ARTICLE DETAILS

TITLE (PROVISIONAL)	Social and structural determinants of injecting-related bacterial and fungal infections among people who inject drugs: protocol for a mixed studies systematic review
AUTHORS	Brothers, Thomas; Lewer, Dan; Bonn, Matthew; Webster, Duncan; Harris, Magdalena

VERSION 1 – REVIEW

REVIEWER	Buchanan , Ashley University of Rhode Island, Pharmacy Practice
REVIEW RETURNED	01-Apr-2021

GENERAL COMMENTS	Review of “Social and structural determinants of injecting-related bacterial and fungal infections among people who inject drugs: protocol for a mixed studies systematic review” General Comments Overall, this is well-written paper about a protocol for a systematic review study to better understand social and structural determinants of injection-related infections among people who inject drugs (PWID). The authors provide strong motivation for this research question and a detailed description of their systematic review procedures. I commend the authors for assembling a research team comprised of people who inject drugs. Their paper and study could be improved by perhaps narrowing the scope of the population(s) studied, consider performing a meta-analysis to synthesize information across quantitative studies, and provide additional context for this study in the presence of other infectious diseases, such as COVID-19, HCV, and HIV/AIDS. Please find a summary of major comments and detailed minor comments below. Specific Comments for Revision Major Comments I commend the authors for assembling a research team comprised of people who inject drugs (current or past). This will bring invaluable knowledge and expertise to this study. The authors may want to consider narrowing the scope of their study to certain age groups and geographic settings. Health outcomes and social/structural factors among PWID may vary meaningfully across different cultural settings. How will the single study proposed here be able to capture and describe these differences? Do the structural and social factors differ depending on the substance used?
---

	The authors should add the dates their study will be conducted and the data ranges for their literature review. How will this study synthesize the information on different exposures and outcomes, if measured differently in the published studies? Given the authors are collecting information on quantitative studies, would it be possible to perform a meta-analysis? Do the authors know how many publications will be included in their study? Were any previous meta-analyses conducted on this topic? To further motivate this paper, the authors could discuss that HIV is on the rise in the US among PWID and discuss the possible connection between these infections and HIV/HCV transmission. Furthermore, what role does COVID have in studying this problem, particularly with access to harm reduction services? The authors may want to discuss how network-based studies of PWID has a role in understanding the social and structural risk factors for injection-related infections. When discussing criminal measures, the authors should also discuss the role of the war on drugs, particularly in US settings. What could be done to address the social and structural causes that might be identified in this study (See Friedman, et al, 2020)? In the administrative claims data for future work, how will social informatics be applied to this context? References: Friedman, S. R., Krawczyk, N., Perlman, D. C., Mateu-Gelabert, P., Ompad, D. C., Hamilton, L., ... & Cerdá, M. (2020). The opioid/overdose crisis as a dialectics of pain, despair, and one-sided struggle. Frontiers in Public Health, 8. Friedman, S. R., Neaigus, A., Jose, B., Curtis, R., & Des Jarlais, D. (1998). Networks and HIV risk: an introduction to social network analysis for harm reductionists. International Journal of Drug Policy, 9(6), 461-469. Minor Comments 1. Introduction: a. In the Introduction, the authors could provide more context for the social and structural factors and the relationships to the outcomes of interest. 2. Methods: a. Page 8, line 55: Will the qualitative studies have analyses of associations? Could the authors clarify what they mean here? b. Page 10, line 32: Please clarify what is meant by individual and composite. c. Page 14, line 21: How will a final decision be reached for discrepancies? d. Page 14, line 41: How will the authors use the information from low quality studies in their work and how will this inform interpretation of the data? Why not exclude these studies? e. Page 15, line 32: For the mixed methods appraisal tool, where would cohort and case-control studies be considered?
--	---

REVIEWER	Chang, Hsing-Yi National Health Research Institutes, Institute of Population Health Sciences
REVIEW RETURNED	05-Apr-2021

GENERAL COMMENTS	This was a protocol for reviewing social and structural determinants of injecting-related to infection in people who inject drugs (PWID). This was an ambitious plan. The only concern was that researchers defined their social and structural determinants differently. Finding a way to combine them would be of challenge. However, authors did mention using qualitative approach for the study.
---

VERSION 1 – AUTHOR RESPONSE

Reviewer 1 Comments (Dr. Buchanan):

Major comments

Comment 3: I commend the authors for assembling a research team comprised of people who inject drugs (current or past). This will bring invaluable knowledge and expertise to this study.

Authors' response: Thank you for this comment, and thank you for your close reading and deep engagement with our manuscript.

Comment 4: The authors may want to consider narrowing the scope of their study to certain age groups and geographic settings. Health outcomes and social/structural factors among PWID may vary meaningfully across different cultural settings. How will the single study proposed here be able to capture and describe these differences? Do the structural and social factors differ depending on the substance used?

Authors' response: Thank you for your close reading of our manuscript and for engaging with our proposed conceptual framework. We recognize that this mixed studies systematic review has a broad scope in terms of the potential exposures (social-structural factors) and potential outcomes (incidence of injecting-related bacterial and fungal infections, or health outcomes afterwards). As this is the first evidence synthesis project on this topic, we have aimed to cast a wide net and capture all relevant evidence.

We agree, as you thoughtfully point out, that health outcomes and social/structural factors vary across different cultural and geographic settings. Your questions posed here are exactly the kinds of questions we seek to answer with this project. If the evidence exists, these differing cultural factors would be included within the social category of Rhodes' "risk environment" framework (which is comprised of social, economic, political, or physical factors). Similarly, the availability of specific substances in a local drug market (a physical, micro-environmental factor) is likely influenced by other micro- and macro-environmental factors like police crackdowns, state drug policies, and local prescribing practices. We are seeking to understand important elements of the risk environment for injecting-related bacterial and fungal infections, rather than determine exactly what is in the risk

environment for any specific individual or sub-population of PWID. We expect to find that micro- and macro-environmental factors influence risk for injecting-related bacterial and fungal infections in different ways in different settings, and this will be the focus of our qualitative content analysis.

Comment 5: The authors should add the dates their study will be conducted and the date ranges for their literature review.

Authors' response: In response to this suggestion and the Editors' suggestion, above, we have added the planned dates for our study and the date ranges for the systematic review.

- Lines 145-146, Methods and analysis:

"We plan to conduct this study for 12 months following the initial full database search on February 18, 2021."

- Lines 242-243, Time frame, setting, and language:

"We will include studies published between January 1, 2000, and the planned search date, February 18, 2021..."

Comment 6: How will this study synthesize the information on different exposures and outcomes, if measured differently in the published studies?

Authors' response: We have conceptualized injecting-related bacterial and fungal infections among people who inject drugs as sharing a common causal pathway, pathophysiology, and potential risk reduction interventions. We expect to identify several different exposures (e.g. homelessness, poverty, sex work, etc.) and several different outcomes (e.g. cellulitis, epidural abscess, death after hospitalization with endocarditis), along a potential pathway from (a) drug acquisition; (b) drug preparation; (c) drug injection; (d) development of and care for superficial infections (e.g. self-treatment or primary/emergency department care); (e) development of and care for severe or invasive infections (e.g. self-care or hospital care); and (f) outcomes after infection or hospital discharge, including infection-related mortality.

For our qualitative, directed content analysis, identifying different exposure and outcome categories/definitions will be important data and a major focus of the synthesis. For a potential quantitative meta-analysis (see response to Comment 7, below), if we identify sufficient quantitative data we would plan to combine multiple infection types for our primary analysis.

- Lines 359-366, Data synthesis and analysis:

"If we identify sufficient quantitative data, we will conduct a meta-analysis of summary effects of association between social/structural factors (e.g. homelessness) and injecting-related infection outcomes among PWID (e.g. hospitalization with any severe injecting-related bacterial or fungal infection) identified in our content analysis. Where available, we will extract measures of association adjusted for confounding, including point estimates and standard errors. To assess feasibility, we will evaluate clinical and methodological heterogeneity by comparing study design, sample characteristics, and operational definitions of exposures and outcome [65,77]. We will measure statistical variability between studies using I^2 statistics.[65,77,78]"

Comment 7: Given the authors are collecting information on quantitative studies, would it be possible to perform a meta-analysis? Do the authors know how many publications will be included in their study? Were any previous meta-analyses conducted on this topic?

Authors' response: To our knowledge no previous meta-analysis has been published on this topic, and we do not know how many publications will be included in the end. Learning from other studies that have performed quantitative meta-analysis on observational studies of social-structural factors (e.g. Gicquelais et al./Reference #65; Platt et al./Reference #78), we have added a statement explaining how we could do so if we identify sufficient quantitative data (see comment #6, above).

Comment 8: To further motivate this paper, the authors could discuss that HIV is on the rise in the US among PWID and discuss the possible connection between these infections and HIV/HCV transmission. Furthermore, what role does COVID have in studying this problem, particularly with access to harm reduction services?

Authors' response: We agree this is a motivational point, to highlight that all injecting-related infections (including viral and bacterial/fungal) share many social-structural associations between them, and that all sorts of injecting-related infections are increasing in incidence in the context of the North American overdose epidemic. The COVID pandemic has had a large impact on people who use drugs and harm reduction services.

In response to this suggestion and Comments 9 and 10, below, we have added further discussion to the "Ethics and dissemination" section. We highlight connections between our proposed approach and rise in HIV incidence and the potential impact of the COVID-19 pandemic:

- Lines 398-405, Ethics and dissemination:

"Incidence of HIV and HCV infections are increasing in parallel with injecting-related bacterial and fungal infections among PWID, in the context of the North American overdose crisis; we expect to find that these outcomes share many overlapping social-structural correlates.[29,45,79,85] Secular increases in drug-related infections and overdoses over the past decade have been describes as a "syndemic", with shared, fundamental causes rooted in the "War on Drugs" and austerity politics.[29,42,79,85] Accordingly, findings from our study may be able to inform risk reduction interventions for multiple health outcomes among PWID."

- Lines 405-408, Ethics and dissemination:

"The COVID-19 pandemic has accelerated the overdose death crisis, deepened poverty and marginalization, and interrupted access to harm reduction services[79,85] – the consequential impacts on injecting-related bacterial and fungal infections, and potential for specific interventions, is not yet clear."

Comment 9: The authors may want to discuss how network-based studies of PWID has a role in understanding the social and structural risk factors for injection-related infections. When discussing criminal measures, the authors should also discuss the role of the war on drugs, particularly in US settings.

Authors' response: Thank you for sharing these references and this perspective on the value of network-based studies, which we had not previously considered in the context of our review and have now added. We have added a discussion on the role of network-based studies in understanding social-structural risk production for bacterial and fungal infections, primarily related to exposure to specific pathogenic bacteria through skin colonization or a sared, contaminated drug supply. We have also added specific discussion of the War on Drugs, in the North American context.

- Lines 416-422, Ethics and dissemination

“Extending further concepts from the HIV prevention social science literature, we are also interested in how our findings may relate to interventions in social networks.[92] Social relationships within networks of PWID influence the likelihood of individual risk or protective behaviours (e.g. needle re-use; skin cleaning). While bacterial and fungal infections are generally not transmitted through blood, as is HIV or HCV, PWID in the same network may experience transmission of colonization with pathogenic bacteria (e.g. MRSA)[93] or heightened risk of accessing a shared, contaminated drug supply (e.g. outbreaks of botulism[3,94] or anthrax[95]).”

- Lines 401-403, Ethics and dissemination

“Secular increases in drug-related infections and overdoses over the past decade have been describes as a “syndemic”, with shared, fundamental causes rooted in the “War on Drugs” and austerity politics.[29,42,79,85]”

Comment 10: What could be done to address the social and structural causes that might be identified in this study (See Friedman, et al, 2020)? In the administrative claims data for future work, how will social informatics be applied to this context?

References:

Friedman, S. R., Krawczyk, N., Perlman, D. C., Mateu-Gelabert, P., Ompad, D. C., Hamilton, L., ... & Cerdá, M. (2020). The opioid/overdose crisis as a dialectics of pain, despair, and one-sided struggle. *Frontiers in Public Health*, 8.

Friedman, S. R., Neaigus, A., Jose, B., Curtis, R., & Des Jarlais, D. (1998). Networks and HIV risk: an introduction to social network analysis for harm reductionists. *International Journal of Drug Policy*, 9(6), 461-469.

Authors' response: In response to this comment, your suggested references, and your several other helpful comments above, we have expanded our discussion on how this study might inform potential future interventions, including “structural interventions”. We have removed specific reference to the emerging field of “social informatics”, and instead simply mention the importance of including consideration of social factors when analyzing linked administrative data.

- Lines 381-391, Ethics and dissemination

“By mapping the risk environment for injecting-related bacterial and fungal infections among PWID, this study aims to identify opportunities for new policy and practice approaches to prevention. This may include highlighting the importance of scaling up access to existing interventions (e.g. needle/syringe distribution; opioid agonist treatment) or identifying opportunities for novel combined social and clinical interventions (e.g. providing Housing First along with antibiotics and addiction treatment for PWID hospitalized with invasive infctions). Our evidence synthesis may identify opportunities for “structural interventions”, which promote health by altering the social-structural context which influence health.[46,85] Examples include the potential impacts of policy or law changes (e.g. decriminalizing substance use may combat stigma and facilitate access to primary care of injecting-related infections) or of community mobilization (e.g. organized PWID empowered to effectively demand safe and welcoming hospital care).[44,86,87]”

- Lines 423-425, Ethics and dissemination

“By emphasizing the potential importance of social and structural factors, the findings will also inform future epidemiological research using large-scale linked administrative data linking individual and clinical risk factors with information on social factors.[69,96]”

Minor Comments

Comment 11: Introduction: In the Introduction, the authors could provide more context for the social and structural factors and the relationships to the outcomes of interest.

Authors' response: We have provided further context and rationale in the "Introduction", and also in response to the Major comments above, we have added much more discussion and context in the "Ethics and dissemination" section:

- Lines 120-125, Introduction:

"For example, homelessness may constrain an individual's ability to wash their hands or use sterile water for injecting,[42] and policy constraints on needle and syringe programs (NSP) (from criminalisation to reduced operating hours) create a situation in which an individual is more likely to reuse a blunted or contaminated needle. Stigma and criminalization of people who use drugs may keep people away from primary health care, causing superficial bacterial infections to remain untreated and progress to enter the bloodstream."

- Lines 381-391, Ethics and dissemination:

"By mapping the risk environment for injecting-related bacterial and fungal infections among PWID, this study aims to identify opportunities for new policy and practice approaches to prevention. This may include highlighting the importance of scaling up access to existing interventions (e.g. needle/syringe distribution; opioid agonist treatment) or identifying opportunities for novel combined social and clinical interventions (e.g. providing Housing First along with antibiotics and addiction treatment for PWID hospitalized with invasive infections). Our evidence synthesis may identify opportunities for "structural interventions", which promote health by altering the social-structural context which influence health.[46,85] Examples include the potential impacts of policy or law changes (e.g. decriminalizing substance use may combat stigma and facilitate access to primary care of injecting-related infections) or of community mobilization (e.g. organized PWID empowered to effectively demand safe and welcoming hospital care).[44,86,87]

Comment 12: Methods: Page 8, line 55: Will the qualitative studies have analyses of associations? Could the authors clarify what they mean here?

Authors' response: Thank you for catching this. We have updated the language to be more accurate.

- Lines 182-186, Study designs:

"We will include studies measuring quantitative associations between exposures and outcomes of interest (as described below), and studies reporting qualitative data on relationships between experiences of exposures and outcomes. We will exclude case-reports and case series that do not include analyses of association, and we will exclude reviews, commentaries, and editorials, as they do not include original data."

Comment 13: Page 10, line 32: Please clarify what is meant by individual and composite.

Authors' response: Thank you for pointing out this lack of clarity. Whether outcomes are classified as individual (e.g. one diagnosis) or composite (e.g. one of multiple possible outcomes) won't actually meaningfully affect our analysis. The main point we wish to make is that we will extract outcomes as reported. We have made this change to remove this ambiguity.

- Lines 236, Outcomes:
“We will extract all outcomes as reported.”

Comment 14: Page 14, line 21: How will a final decision be reached for discrepancies?

Authors' response: Clarification added:

- Lines 335-337, Data synthesis and analysis:
“To improve reliability, coding will be performed in tandem by a reviewer pair for three studies, with discrepancies resolved through discussion.”

Comment 15: Page 14, line 41: How will the authors use the information from low quality studies in their work and how will this inform interpretation of the data? Why not exclude these studies?

Authors' response: We plan to use the systematic quality assessment primarily to understand the quality of the literature in different areas as we map the risk environment. We will be able to report the graded quality of each study included in our synthesis. We are choosing not to exclude these studies a priori because we wish to understand full existing research literature, which we believe to be relatively limited.

Comment 16: Page 15, line 32: For the mixed methods appraisal tool, where would cohort and case-control studies be considered?

Authors' response: For the mixed methods appraisal tool, case-control studies would be considered within the “nonrandomized controlled” category, and cohort studies would be included within the “quantitative descriptive” category.

Reviewer 2 comments (Dr. Chang):

Comment 17: This was an ambitious plan. The only concern was that researchers defined their social and structural determinants differently. Finding a way to combine them would be of challenge. However, authors did mention using qualitative approach for the study.

Authors' response: Thank you very much. We do plan to use our qualitative content analysis to understand and describe the way social structures influence or create risk of injecting-related bacterial and fungal infections among people who inject drugs.

We are grateful for your consideration of this paper and feel your suggestions have improved it significantly. We look forward to hearing from you regarding the disposition of the manuscript.